# Physiotherapist' job performance, impression management and organizational citizenship behaviors: An analysis of hierarchical linear modeling

Fu-I Hou[1][¤a], Yu-Lung Wu[2][¤a], Min-Hui Li[3,4][¤b], Wan-Yun Huang[1,3,5][¤b]*

1 Department of Information Engineering, I-Shou University, Kaohsiung, Taiwan, 2 Department of Information Management, I-Shou University, Kaohsiung, Taiwan, 3 Department of Physical Medicine and Rehabilitation, Kaohsiung Veterans General Hospital, Kaohsiung, Taiwan, 4 Graduate Institute of Aerospace and Undersea Medicine, National Defense Medical Center, Taipei, Taiwan, 5 Institute of Allied Health Sciences, National Cheng Kung University, Tainan, Taiwan

¤a Current address: Dashu District, Kaohsiung City, Taiwan, R.O.C
¤b Current address: Zuoying Dist., Kaohsiung City, Taiwan, R.O.C
* ballan666888@gmail.com

**Data Availability Statement:** All relevant data are within the manuscript and its Supporting Information files. The protocol is available from

## Abstract

Studies on physiotherapists are generally focused on clinical professionalism, with very few examining job performance from a management standpoint. To address this gap, this study sought to investigate the relationship between impression management and organizational citizenship behavior and job performance. This study targeted medical institutions offering rehabilitation and physiotherapy services and conducted a questionnaire survey based on scales developed by domestic and foreign scholars. A total of 600 questionnaires were distributed and 523 valid ones collected. The data was tested and verified using regression analysis and hierarchical linear modeling (HLM). In the survey, the Impression Management Scale, Organizational Citizenship Behavior Scale, and Job Performance Scale indicated that at the individual level, the impression management of physiotherapists is significantly related to their organizational citizenship behaviors and job performance. The organizational citizenship behaviors were also found to have a mediating effect between impression management and job performance. At the group level, impression management had a conditioning effect on organizational citizenship behaviors and job performance. In terms of statistical methods, group-level variables act as moderators, which affects the power of individual-level explanatory variables on outcome variables, i.e., the influence of the slope. The job behaviors of physiotherapists entail direct service and their performance is closely related to organizational development. Impression management gives people certain purposes and behaviors while organizational citizenship behaviors are a type of non-self-seeking, selfless dedication behaviors. Therefore, the motivation of physiotherapists who demonstrate organizational citizenship behaviors should be further explored.

Protocols.io (DOI: 10.17504/protocols.io.
bq2amyae).

**Funding:** This work was supported by the study
and research funding of Kaohsiung Veterans
General Hospital (Project VGHKS106-036).

**Competing interests:** NO authors have competing
interests.

# Introduction

Current medical environments are patient-centered, and rehabilitation medical services and
needs have significantly increased [1, 2]. Moreover, in an era of comprehensive competition,
the quality of medical institutions is an important strategic indicator of competitiveness in the
industry [3]. In medical management, it is necessary to comprehensively understand the fac-
tors that affect the work attitude of physiotherapists to improve the quality of services. There-
fore, apart from overall organizational operations, monitoring employee behavior is also
important for hospitals. Employees motivated by impression management will use organiza-
tional citizenship behaviors to influence others' perceptions to increase their personal appeal
and achieve their desired outcome [4, 5]. Furthermore, employee behaviors become goal-ori-
ented and outward conduct becomes intentional to achieve the desired effect [6]. On the con-
trary, when goals are personally favorable, employees grow more diligent in impression
management [7]. To earn promotions and pay raises, employees often adopt certain strategies
to influence others and achieve their goals through impression management [8–12]. People
aim to appear competent to others and, thus, demonstrate diverse positive impressions in
many areas of life [13, 14].

Organizational citizenship behaviors are not clearly defined in organizational systems and
not mandated in employees' basic job requirements and manuals [15, 16]. Sun et al. [17]
emphasized that enhancing organizational citizenship behaviors significantly associates team
and job performance. Mcshane and Glinow point out that these behaviors include active partici-
pating in organizational activities, selflessly assisting colleagues, and reducing unnecessary con-
flicts [4]. Konovsky and Pugh opine that the performance of organizational citizenship
behaviors is not subject to the requirements of the employee's task and the ability to judge their
performance; they also claim that employees do not mind whether they will be rewarded [18].

Gaes and Tedeschi [19] hold that the pursued values and goals affect employee motivation
in impression management. Bolino [20] identifies several similarities between the concepts of
impression management and organizational citizenship, such as actively helping colleagues. In
impression management, individuals aim to influence decision makers' behaviors by increas-
ing their attractiveness to others [21]. Similarly, organizational citizenship behaviors are the
most direct and strategic tactic for individuals to reciprocate to their colleagues by unselfishly
helping and cooperating with them to ensure high job performance [22].

Robbins and Judge [23, 24] view the impression management strategy as the displayed
behavior before making a request. In other words, it refers to the way one treats work perfor-
mance as a professional and how actively they participate in their work, and what they think is
important to them in their work. Mcshane and Glinow assert that individuals create favorable
impressions and help their colleagues at work gain approval from others [4]. In a particular
workplace, when employees share a common understanding of the factors affecting their work
environment, their shared perception combines to form a specific construct [25]. This con-
struct shares the same content, meaning, and constructive validity in a multilevel aggregate of
data [26], making construct theories highly valuable in multilevel studies.

Vroom [27] defines job performance as an achievement resulting from the interaction
between motivation and ability. Katz and Kahn [28], on the other hand, view job performance
as job standards and behaviors defined by an organization. Performance results from the inter-
action of factors such as employee effort, capability, and role perception in a particular circum-
stance [29, 30]. It produces value in a job outcome through behavior and the use of
testimonies of that value to adjust expectations of job output and behavior [31].

Gaes and Tedeschi [19] believe that values and goals affect employee motivation in impres-
sion management. Bolino [20] suggests many similarities between the concepts of impression

management and organizational citizenship, such as actively helping colleagues. Impression management is the individuals' aim to increase their attractiveness to others and thereby influence decision makers' behaviors [21]. Organizational citizenship is one of the most direct and visible strategies where individuals rely on their colleagues to help achieve important work goals [22].

Physiotherapists serve the patient directly. While behavioral performance affects the development of the overall organization, impression management is purposeful and engaged behavior, and organizational citizenship behavior an act of selfless dedication to the self. When practitioners of impression management demonstrate organizational citizenship behaviors, their motivation becomes a question that can be explored. Therefore, this study seeks to examine the effects of physiotherapists' impression management, study whether organizational citizenship behavior affects their job performance, and identify the relationship between various aggregates.

The purposes of this study are as follows: 1) To investigate the relationship between physiotherapists' impression management and organizational citizenship behavior; 2) to explore the relationship between physiotherapists' organizational citizenship behavior and job performance; 3) to inspect whether physiotherapists' impression management will affect job performance through organizational citizenship behavior; and 4) to make recommendations based on the results for reference by medical institutions.

Hence, this study hypothesized the following:

Hypothesis 1: Impression management positively associates organizational citizenship behaviors (H1);

Hypothesis 2: Organizational citizenship behaviors positively associate job performance (H2);

Hypothesis 3: Organizational citizenship has a mediating effect between impression management and job performance (H3);

Hypothesis 4: Aggregate impression management positively associates organizational citizenship behaviors (H4);

Hypothesis 5: Aggregate impression management positively associates job performance (H5); and

Hypothesis 6: Aggregate impression management has a conditioning effect on organizational citizenship and job performance (H6).

The above hypothetical deductions are summarized in Fig 1.

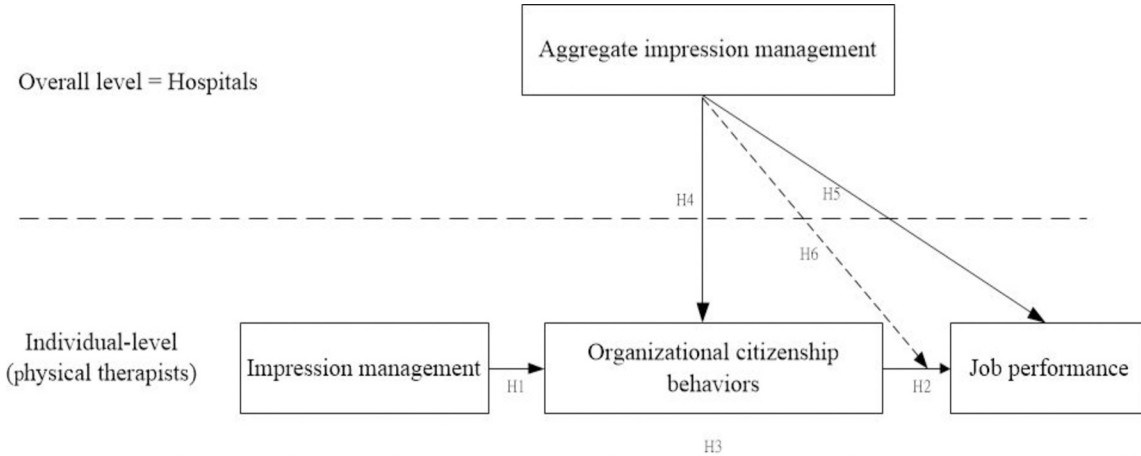

**Fig 1. Research framework.**

## Materials and methods

A cross-sectional design was employed in the study. All study subjects provided consent after being given sufficient information. The study was approved by the Institutional Review Board of the Kaohsiung Veterans General Hospital, Taiwan (IRB No.: VGHKS16-CT9-2).

This research ensured avoiding the problems of homologous bias and common method variation in the collection of data. It is considered that the collection and measurement of performance indicators should be recommended after the high-performance work system [32]. The questionnaires used in this study were distributed in two stages. The first stage was to measure the variables of the high-performance work system. The second questionnaire was executed two months after the first was issued. Since this research adopted a cross-layer analysis, the unit of the second questionnaire was primarily the unit that was issued the first time and the professional performance was carried out.

In the questionnaire survey, the scales for impression management, organizational citizenship behaviors, and job performance were adopted from Kumar and Beyerlein [33], Polsakoff et al. [34], and Motowidlo and Van Scotter [35], respectively. The responses are measured on a five-point Likert scale, from largely disagree to largely agree, with 1 to 5 points. The correlation coefficient between impression management and organizational citizenship behaviors was 0.417, between organizational citizenship behaviors and job performance was 0.679, and between impression management and job performance was 0.204. All values showed a significant positive correlation and reliability above 0.7 (Table 1).

### Statistical analysis

The relevant analysis and reliability tests were conducted on the sample using the SPSS20 statistical software for Windows. The hierarchical linear modeling (HLM) was then constructed using HLM 7.01 for regression analysis.

The analysis included (1) a null model for calculating the intra-class correlation coefficient (ICC) to determine differences among the overall levels and (2) intercepts- and slope-as-outcome for exploring the impact of all independent variables on the effect of dependent variables at the individual level and using the overall level independent variables to explain the variability of dependent variable at the individual level.

## Results

This study distributed 600 questionnaires to practicing physiotherapists in Taiwan, Penghu, and Kinmen. A total of 543 questionnaires were collected, of which 523 were valid, representing an effective recovery rate of 87.2%. Table 2 shows the sample distribution.

**Table 1. Descriptive statistics, correlation coefficients, and reliability coefficient of the research variables.**

| Research variables | M | SD | Cronbach, α | Research variables' related data | | |
|---|---|---|---|---|---|---|
| | | | | (1) | (2) | (3) |
| **Impression management** | 2.84 | 0.61 | 0.93 | 1 | | |
| **Organizational citizenship behaviors** | 3.29 | 0.35 | 0.75 | 0.417*** | 1 | |
| **Job performance** | 3.80 | 0.46 | 0.95 | 0.204*** | 0.679*** | 1 |

*$p < .05$

**$p < .01$, and

***$p < .001$.

**Table 2. Description of the sample (n = 523).**

|  | Category | Participants | % |
|---|---|---|---|
| **Physiotherapist** |  | 523 | 100 |
| **SEX** | Male | 215 | 41.1 |
|  | Female | 308 | 58.9 |
| **Office unit** | Medical center | 43 | 8.22 |
|  | Regional hospital | 73 | 13.96 |
|  | Local education | 130 | 24.86 |
|  | Clinic | 223 | 42.64 |
|  | Others | 54 | 10.32 |

The regression analysis revealed that the four dimensions of self-demonstration in impression management had a significant positive impact on organizational citizenship behaviors with an explanatory power of 19.4% and an F-value of 32.318 (Table 3). Hence, H1 is partially supported.

Other levels exhibited a significant positive impact on job performance, except for sportsmanship, with an explanatory power of 60.3% and a significant F-value of 157.085 (Table 4). Hence, H2 is partially supported.

Brownnosing to understand the intentions of superiors and self-demonstration showed a significant impact on job performance (Model 1), with $R^2 = 0.105$, an explanatory power of 10.5%, and a significant F-value of 15.198. The independent variables in Model 2 showed that among the four dimensions of impression management, only brownnosing displays a significant positive impact on job performance after the mediating effect of organizational citizenship behaviors. Among the dependent variables in Model 2, the diligence, civil ethics, humility and courtesy, and altruism variables showed a significant impact on job performance with an explanatory power of 62% and a significant F-value of 138.896 (Table 5). Hence, H3 is supported.

## Hierarchical linear modeling

The null model was first used for the significance test of the intergroup variance to determine whether HLM is appropriate for the study data. Table 6 shows that $\tau00 = 0.027$ and $p < 0.001$, which demonstrates a significant estimate of intergroup variance. Therefore, a basis for testing group explanatory variables and individual explanatory variables for variance within organizational citizenship behaviors is available. In the HLM analysis, the null model first calculated the ICC, where ICC = $\tau00 / (\tau00 + \sigma2) = 20.027 / 0.027 + 0.215 = 0.027 / 0.242 = 0.11157$ (p-

**Table 3. Regression analysis of impression management and organizational citizenship behaviors.**

| Criterion<br>Predictor | Organizational citizenship behaviors |
|---|---|
| Ingratiation | .103*** |
| Opinion conformity | −.013*** |
| Rendering Favors | .058*** |
| Self-presentation | .346*** |
| F | 32.318*** |
| $R^2$ | .194 |

***$p < .001$.

**Table 4. Regression analysis of organizational citizenship behaviors and job performance.**

| Criterion / Predictor | Job performance |
|---|---|
| Conscientiousness | .149*** |
| Sportsmanship | −.301 |
| Civic virtue | .121*** |
| Courtesy | .299*** |
| Altruism | .369*** |
| F | 157.085*** |
| $R^2$ | .603 |

Note: *$p < .05$

**$p < .01$, and

***$p < .001$.

value 0.000). Cohen [36] suggests that for moderate correlation, a variance of 0.027 due to differences among hospitals should account for approximately 11.1% of the variance. The ICC1 estimation results indicated that the dependent variable of job performance presented group differences. Thus, the characteristics of intergroup differences must be considered. In the intercept- and slopes-as-outcome model, single or multiple continuous independent variables were placed in the second level (overall level) as the explanatory variable for the first level intercept (e.g., Yj, the average subgroup value of Y) to indirectly explain Y. However, for the individual level, the explanatory variables have no predictive model. Therefore, to explain the variance for the level 1 intercept, this study estimated an HLM model for organizational citizenship behaviors and job performance, where the aggregate impression management in the level 1 equation was used as the explanatory variable in the level 2 equation. The aggregate impression management ($\gamma 01 = 0.037$ ($p < .05$) displayed a predominant effect on organizational citizenship behaviors and impression management ($\gamma 01 = 0.172$ ($p < .05$), and a cross-level effect on job performance, thereby supporting H4 and H5.

**Table 5. Regression analysis of impression management and organizational citizenship behaviors on job performance.**

| Invest in the variable | Model 1(β) | Model 2(β) |
|---|---|---|
| Ingratiation | .230*** | .163*** |
| Opinion conformity | −.324*** | −.104* |
| Rendering Favors | .101** | .005 |
| Self-presentation | .210*** | −.098 |
| Conscientiousness | | .163*** |
| Sportsmanship | | −.016 |
| Civic virtue | | .127*** |
| Courtesy | | .268*** |
| Altruism | | .398*** |
| F | 15.198*** | 92.910*** |
| △F | 15.198*** | 138.896*** |
| $R^2$ | .105 | .620 |
| △$R^2$ | .098 | .613 |

Note: *$p < .05$

**$p < .01$, and

***$p < .001$.

**Table 6. Hierarchical linear model of the various indicators.**

|  | γ00 | γ01 | γ10 | γ20 | τ00 | τ11 | σ² | deviance |
|---|---|---|---|---|---|---|---|---|
| 1. Null model | 3.79*** (0.021) |  |  |  | 0.027*** |  | 0.215 | 686.222 |
| L1: Y(JP) = β0j + rij |  |  |  |  |  |  |  |  |
| L2: β0j = γ00 + U0j |  |  |  |  |  |  |  |  |
| 2. Intercepts-as-outcomes models | 3.76*** (0.29) | 0.172* (0.09) |  |  | 0.013* |  | 0.21 | 684.708 |
| (1) AIM–JP |  |  |  |  |  |  |  |  |
| L1: Y(JP) = β0j + rij |  |  |  |  |  |  |  |  |
| L2: β0j = γ00 + γ01(AIM) + U0j |  |  |  |  |  |  |  |  |
| (2)AIM–OCB–JP | 0.74*** (0.173) | 0.037* (0.068) | 0.912*** (0.051) |  | 0.11* | 0.09* | 0.11 | 337.839 |
| L1: Y(JP) = β0j + β1(OCB) + rij |  |  |  |  |  |  |  |  |
| L2: β0j = γ00 + γ01(AIM) + U0j |  |  |  |  |  |  |  |  |
| β1i = γ10 + U1i |  |  |  |  |  |  |  |  |
| 3. Slopes-as-outcomes model | 0.816*** (0.18) | 0.11* (0.08) | 0.08** (0.027) | 0.96*** (0.05) | 0.09* | 0.02 | 0.1 | 329.604 |
| (1) AIM–OCB–JP |  |  |  |  |  |  |  |  |
| L1: Y(JP) = β0j + β1(OCB) + β2 (IM) + rij |  |  |  |  |  |  |  |  |
| L2: β0j = γ00 + γ01 (AIM) + U0j |  |  |  |  |  |  |  |  |
| β1i = γ10 + γ11 (AIM) + U1i |  |  |  |  |  |  |  |  |

Note: L1 = Individual-level; L2 = Overall level; IM = impression management; AIM = aggregate impression management; OCB = organizational citizenship behaviors; JP = job performance

### Cross-level conditioning effect: Slope prediction model

Both the first and second levels had one or more continuous independent variables as explanatory variables for Y and the regression coefficients, respectively. Thus, a complete model integrating the slope and intercept models is achieved and referred to as the slope prediction model. Group explanatory variables were used to determine the explanatory power of the variance. As shown in Table 6, the estimated value of the parameters of γ11 achieved significant levels. The relationship between organizational citizenship behaviors and job performance indicates that aggregated impression management showed a significant slope prediction, where γ10 = 0.08 (p < 0.5), thereby supporting H6.

## Discussion

This study used HLM to analyze the cross-level relationship between individual-level impression management and organizational citizenship behaviors and group-level hospital aggregate impression management variables. The regression analysis showed that impression management had a significant positive effect on organizational citizenship behaviors and job performance, with organizational citizenship behaviors exhibiting a mediating effect, similar to Lin's findings [37].

The HLM analysis supported the cross-level conditioning effect of impression management, which suggests that when physiotherapists had greater impression management, they were more associates to affect organizational citizenship behaviors to change the perception of others toward them, hence increasing their chance of achieving their goals. In other words, physiotherapists used positive behaviors such as creating goodwill among peers by helping others and working hard to impress their supervisors, thus creating favorable impressions.

There is an important assumption in the multi-level model analysis that may affect the interpretation of individual-level variables for outcome variables, i.e., multi-level models have cross-level interactions [38]. In terms of statistical methods, group-level variables act as

moderators, which affects the power of individual-level explanatory variables on outcome variables, i.e., the influence of the slope [39].

This study found that the physiotherapists achieved their interests by building friendly relationships, actively participating in organizational events, and displaying an interest in organizational development. Their impression management brought about two effects. The greater their motivation to impress their superiors, the weaker their demonstration of organizational citizenship behaviors—owing to the fact that the most direct way to receive reciprocation from the organization is to target superiors. Correspondingly, the motivation for establishing a favorable role by relying on organizational citizenship behaviors relatively weakened, similar to Lin's findings [37].

When physiotherapists focused their attention exclusively on their superiors, they became counter-productive. Although impression management benefits the organization, it can be damaging. In addition to catering to their supervisors, organization members may also be dedicated to their jobs to further impress their supervisors, thereby gaining their desired rewards. Therefore, hospital managers should balance various considerations before making decisions.

Thus, physiotherapists' attitudes are also predictors of medical quality. Their benign interaction with each other helps improve their job attitudes, which allows patients to receive quality care and increases their loyalty to the treatment team and referral rate to other patients. As a result, a virtuous circle is created and hospital performance improved.

This study explored the mediating effect of organizational citizenship behaviors between impression management and job performance. Most physiotherapists demonstrated diligence in their duties, compliance with civil ethics, humility and courtesy in their service, and altruism, impacting their clinical job performance positively. At the hospital level, impression management primarily impacted the treatment service behaviors and job performance of the physiotherapists. Thus, further considerations must be given to establishing guidelines that are helpful to hospital impression management and professional behaviors of the physiotherapists. Future research should include more hospital-level factors to examine the relationship between organizational citizenship behaviors and job performance. This approach provides a reference to superiors in managing their physiotherapists to better improve medical quality and efficiency. In conclusion, establishing effective management policies in physiotherapists' work environment can help ensure medical quality and patient safety.

## Study limitations

This study employed a questionnaire survey. Although the results verified most of the hypotheses, complete reflection on actual situations using quantitative research is difficult. Moreover, this study is limited by its sole reliance on cross-sectional observation. Therefore, a more rigorous study is required by using questionnaires and interviews for a more holistic conclusion and longer observation time.

## Clinical messages

Impression management positively associates organizational citizenship behaviors. Consequently, organizational citizenship behaviors positively impact job performance and have a mediating effect between impression management and job performance.

Physiotherapists' job behaviors entail direct service, with their performance being closely tied to organizational development. Impression management gives people certain purposes and behaviors and organizational citizenship behaviors are a type of non-self-seeking, selfless dedication behaviors.

## Supporting information

**S1 Table. Descriptive statistics, correlation coefficients, and reliability coefficient of the research variables.**
(DOCX)

**S2 Table. Description of the sample (n = 523).**
(DOCX)

**S3 Table. Regression analysis of impression management and organizational citizenship behaviors.**
(DOCX)

**S4 Table. Regression analysis of organizational citizenship behaviors and job performance.**
(DOCX)

**S5 Table. Regression analysis of impression management and organizational citizenship behaviors on job performance.**
(DOCX)

**S6 Table. Hierarchical linear model of the various indicators.**
(DOCX)

**S1 Data.**
(XLSX)

## Acknowledgments

We deeply appreciate all the participants and Thi-Thanh-Nga Nguyen who contributed to this study.

## Author Contributions

**Conceptualization:** Yu-Lung Wu.

**Data curation:** Fu-I Hou, Wan-Yun Huang.

**Formal analysis:** Fu-I Hou, Wan-Yun Huang.

**Methodology:** Yu-Lung Wu.

**Resources:** Min-Hui Li, Wan-Yun Huang.

**Software:** Yu-Lung Wu, Min-Hui Li, Wan-Yun Huang.

**Supervision:** Min-Hui Li.

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
