## [Decision Letter · Decision Letter 0]

9 Nov 2020

PONE-D-20-16522

Physiotherapist’ Job Performance, Impression Management and Organizational Citizenship Behaviors: An Analysis of Hierarchical Linear Modeling

PLOS ONE

Dear Dr. Huang,

Thank you for submitting your manuscript to PLOS ONE. After careful consideration, we feel that it has merit but does not fully meet PLOS ONE’s publication criteria as it currently stands. Therefore, we invite you to submit a revised version of the manuscript that addresses the points raised during the review process.

To be frank there has been much difficulty eliciting an academic editor and reviewers for this submission. I am taking the liberty of sending this back to you now despite only being able to secure one peer reviewer response, despite the passage of time. You will find some useful constructive comments below that can allow you to evolve the reporting of your work.

We look forward to receiving your revised manuscript.

Kind regards,

Shane Patman, PhD

Academic Editor

PLOS ONE

Journal Requirements:

3. Thank you for stating the following in the Sources of Funding Section of your manuscript:

"This work was supported by the study and research funding of Kaohsiung Veterans General Hospital (Project VGHKS106-036)"

"NO - Include this sentence at the end of your"

Reviewers' comments:

Reviewer's Responses to Questions

**Comments to the Author**

1. Is the manuscript technically sound, and do the data support the conclusions?

Reviewer #1: Partly

2. Has the statistical analysis been performed appropriately and rigorously? 

Reviewer #1: No

3. Have the authors made all data underlying the findings in their manuscript fully available?

Reviewer #1: No

4. Is the manuscript presented in an intelligible fashion and written in standard English?

Reviewer #1: No

5. Review Comments to the Author

Reviewer #1: Physiotherapists’ Job Performance, Impression Management and Organizational

Citizenship Behaviors: An Analysis of Hierarchical Linear Modeling

Detailed review

This paper presents a study that examines the relationship between impression management, organizational citizenship behavior and job performance. This paper touches on a timely, important topic. Thus, when I started reading the paper, I felt it could make a real contribution to the literature as well as have practical implications.

That said, I have some concerns, some major, that need to be addressed before the paper can be considered for publication, as well as a few minor ones.

• One of my main concerns is a possible common method bias. Because physiotherapists completed all the measures, how could the authors be sure that common method bias was avoided?

• In the abstract the authors state that "At the group level, impression management had a conditioning effect on organizational citizenship behaviors and job performance." Please clarify in what way.

• In the conclusion to the abstract the authors state that "The job behaviors of physiotherapists entail direct service and their performance is closely related to organizational development."- This sentence is vague and I do not understand the connection between physiotherapists and organizational development. What does it mean that their behavior is closely related to organizational development? In what way? How does physiotherapists' job differ from any other job with respect to organizational development?

• The authors should provide more references supporting their claims. For example even in the beginning of the introduction (p. 5 lines 66-69): "Current medical environments are patient-centered, and rehabilitation medical services and needs have increased significantly (Author? Date ?). Moreover, in an era of comprehensive competition, medical institutions’ quality is an important strategic indicator of competitiveness in the medical industry." (Author? Date?

• The authors state on p. 5 lines 69-71: "In terms of medical management, factors affecting physiotherapists’ attitude should be completely understood to improve

service quality." – how can attitudes be "completely understood" – I suggest using more accurate terms throughout the paper.

The rationale for the study and the hypotheses should be better developed and presented in a more integrative way.

• On p. 6 lines 86-88 the authors state that "They [OCB] are spontaneous behaviors that are not described in job manuals and occur when employees assist each other to accomplish their tasks." – I do not agree that organizational citizenship behavior is only "spontaneous behaviors". Please provide references from the literature on OCB, and in particular supporting this notion of 'spontaneous behaviors' and expand the definition.

• The authors cite references from books (not updated) on organizational behavior (i.e. No. 19) – "Robbins and Judge (2011)." – The authors should conduct a better and more updated literature search to support their arguments.

• On p. 8 lines 124-125 the authors claim that "Therefore, this study wants to understand the

motivation of physiotherapist's impression management" Didn’t the authors want to examine the effects of impression management? This sentence in misleading.

• In their hypotheses the authors use the word "impact". Because they did not use an experimental design they can't deduce causality and thus should only use terms such as "associate" or "predict".

• In the method section the authors should include a specific description of each of their measures according to APA guidelines.

• The results are not presented according to the APA format. Crucially, that authors should present the exact p values. In addition, the reliabilities should come after the measures subsection (not in a Table of means, SD and inter-correlations).

• Again on p. 12 and in the discussion section the authors use the word "impact" although they can't deduce causality.

• In the discussion the authors state that "…physiotherapists had greater impression management, they were more motivated to manipulate organizational citizenship

behaviors to change the perception of others toward them…" – I don’t think that the authors used any measure that can show that physiotherapists manipulated their organizational citizenship behaviors. They authors did not examine this.

• The paper should be sent for English editing.

6. PLOS authors have the option to publish the peer review history of their article (what does this mean?). If published, this will include your full peer review and any attached files.

Reviewer #1: **Yes: **Dr. Erez Yaakobi

---

## [Decision Letter · Decision Letter 1]

5 May 2021

Physiotherapist’ Job Performance, Impression Management and Organizational Citizenship Behaviors: An Analysis of Hierarchical Linear Modeling

PONE-D-20-16522R1

Dear Dr. Huang,

We’re pleased to inform you that your manuscript has been judged scientifically suitable for publication and will be formally accepted for publication once it meets all outstanding technical requirements.

Kind regards,

Shane Patman, PhD

Academic Editor

PLOS ONE

Additional Editor Comments (optional):

Reviewers' comments:

Reviewer's Responses to Questions

**Comments to the Author**

1. If the authors have adequately addressed your comments raised in a previous round of review and you feel that this manuscript is now acceptable for publication, you may indicate that here to bypass the “Comments to the Author” section, enter your conflict of interest statement in the “Confidential to Editor” section, and submit your "Accept" recommendation.

Reviewer #2: All comments have been addressed

2. Is the manuscript technically sound, and do the data support the conclusions?

Reviewer #2: Yes

3. Has the statistical analysis been performed appropriately and rigorously? 

Reviewer #2: Yes

4. Have the authors made all data underlying the findings in their manuscript fully available?

Reviewer #2: Yes

5. Is the manuscript presented in an intelligible fashion and written in standard English?

Reviewer #2: Yes

6. Review Comments to the Author

Reviewer #2: The revisions looks much better the earlier version and the authors have tried to address most of the concerns raised.

7. PLOS authors have the option to publish the peer review history of their article (what does this mean?). If published, this will include your full peer review and any attached files.

Reviewer #2: No

---

## [Editor Report · Acceptance letter]

12 May 2021

PONE-D-20-16522R1 

Physiotherapist’ Job Performance, Impression Management and Organizational Citizenship Behaviors: An Analysis of Hierarchical Linear Modeling 

Dear Dr. Huang:

I'm pleased to inform you that your manuscript has been deemed suitable for publication in PLOS ONE. Congratulations! Your manuscript is now with our production department. 

Kind regards, 

on behalf of

Assoc Prof Shane Patman 

Academic Editor

PLOS ONE